# Hydrodynamics of spin currents

Angel Domingo Gallegos[1], Umut Gürsoy[1] and Amos Yarom[2]

**1** Institute for Theoretical Physics, Utrecht University,
Leuvenlaan 4, 3584 CE Utrecht, The Netherlands
**2** Department of Physics, Technion, Haifa 32000, Israel

## Abstract

We study relativistic hydrodynamics in the presence of a non vanishing spin potential. Using a variety of techniques we carry out an exhaustive analysis, and identify the constitutive relations for the stress tensor and spin current in such a setup, allowing us to write the hydrodynamic equations of motion to second order in derivatives. We then solve the equations of motion in a certain dynamical spin limit and in a perturbative setup and find surprisingly good agreement with measurements of global Λ-hyperon polarization carried out at RHIC.



## 1 Introduction

The hydrodynamic behaviour of a spin current has been playing an increasingly prominent role in a variety of physical systems ranging from heavy ion collisions to condensed matter experiments. In particular, the recent observation of global spin polarization of the Λ and $\bar{\Lambda}$ particles in heavy-ion collisions at RHIC [1,2] and the experimental realization of spin currents

induced by vorticity in liquid metals [3] have aroused strong interest in the subject, calling for a theoretical underpinning of hydrodynamics in the presence of a spin current.

The derivation of a complete and consistent set of constitutive relations in spin hydrodynamics is lacking in the literature. The main goal of this work is to provide the tools for carrying out such an analysis and to use these tools to obtain the constitutive relations for a parity invariant and conformal fluid to subleading order in a derivative expansion.

Recall that hydrodynamics is a universal low energy effective field theory of many body, finite temperature systems. The equations of motion of hydrodynamics consist of local conservation laws (e.g., energy momentum conservation or charge conservation). The dynamical variables are given by a temperature field $T$, a velocity field $u^\mu$ (which we normalize such that $u^\mu u_\mu = -1$ in the relativistic setting we are working in), and chemical potentials associated with other conserved charges present. In the current context angular momentum conservation leads to a (non-)conservation equation for the spin current, which implies the existence of a spin potential $\mu_{ab}$, i.e. the spin analog of electric chemical potential.

For a system where the only conserved charge is the spin current and energy momentum tensor, and in the absence of anomalies, one finds

$$
\begin{aligned}
\mathring{\nabla}_\mu T^{\mu\nu} &= \frac{1}{2} R^{\rho\sigma\nu\lambda} S_{\rho\lambda\sigma} - T_{\rho\sigma} K^{\nu ab} e^\rho{}_a e^\sigma{}_b \,, \\
\mathring{\nabla}_\lambda S^\lambda{}_{\mu\nu} &= 2 T_{[\mu\nu]} - 2 S^\lambda{}_{\rho[\mu} e_{\nu]}{}^a e_\rho{}^b K_{\lambda ab} \,,
\end{aligned}
\tag{1}
$$

where we denote the vielbein by $e^\mu_a$ and will use it to convert spacetime indices to tangent bundle indices. $A_{[\alpha\beta]} = \frac{1}{2}(A_{\alpha\beta} - A_{\beta\alpha})$, $R^\alpha{}_{\beta\gamma\delta}$ is the Riemann tensor and $K_\mu{}^{ab}$ is the contorsion tensor, related to the spin connection, $\omega_\mu{}^{ab}$ via $\omega_\mu{}^{ab} = \mathring{\omega}_\mu{}^{ab} + K_\mu{}^{ab}$ where $\mathring{\omega}_\mu{}^{ab} = e_\nu{}^a \left( \partial_\mu e^{\nu b} + \mathring{\Gamma}^\nu{}_{\sigma\mu} e^{\sigma b} \right)$ with $\mathring{\Gamma}^\alpha{}_{\beta\gamma} = g^{\alpha\delta} \left( \partial_{(\beta} g_{\gamma)\delta} - \frac{1}{2} \partial_\delta g_{\beta\gamma} \right)$. That is, ringed connections and derivatives denote expressions evaluated using the Christoffel connection.

The virtue of using torsion in intermediate stages of the computation is that it allows us to uniquely determine the spin current and stress tensor. Recall that the stress tensor and spin current can be modified via a Belinfante-Rosenfeld transformation [4,5]. This transformation, sometimes referred to as a pseudo gauge transformation, leads to an ambiguity in the spin current and energy momentum tensor. Fortunately, in the presence of torsion this ambiguity is removed, much like the ambiguity in the definition of the stress tensor is removed by defining it via its coupling to an external metric. Of course, the torsion and curvature should be set to zero when considering, e.g., heavy ion collisions.

To obtain the explicit form of the equations of motion for the hydrodynamic variables one needs a set of constitutive relations whereby all the conserved charge densities (including the energy momentum tensor) are expressed in terms of $T$, $u^\mu$, the relevant chemical potentials and their derivatives. These constitutive relations must satisfy certain criteria which have been shown to be captured by the second law of thermodynamics, at least to leading order in a derivative expansion [6–12]. Often, such constitutive relations are expressed in terms of a truncated expansion in derivatives of the hydrodynamic variables. As we will discuss at length shortly, an unusual feature of hydrodynamics with a spin current is that the spin potential is naturally associated with terms which are first order in derivatives.

In this work we compute the constitutive relations for the stress tensor $T^{\mu\nu}$ and spin current $S^\mu{}_{\alpha\beta}$ of a parity invariant conformal theory in $3+1$ dimensions, in a flat, torsionless background geometry, including all terms which contribute to the equations of motion expanded to second order in derivatives. Restricting these to what we refer to as the dynamical spin limit, we find

$$T^{(\mu\nu)} = \epsilon_0 T^4 u^\mu u^\nu + \frac{1}{3}\epsilon_0 T^4 \Delta^{\mu\nu} - 2\eta_0 T^3 \sigma^{\mu\nu} + T_{BR}^{(\mu\nu)} + \mathcal{O}(\partial^2)$$

$$
\begin{aligned}
T^{-2}T^{[\mu\nu]} =& \Delta_\beta^{[\mu}u^{\nu]}\left(\ell_1 \mathcal{D}_\alpha \sigma^{\alpha\beta} + \ell_2 \mathcal{D}_\alpha \hat{M}^{\alpha\beta}\right) + \ell_3 \Delta^{\rho[\mu}\Delta^{\nu]\sigma}\mathring{\nabla}_\rho \hat{m}_\sigma + \ell_4 u^{[\mu}\sigma^{\nu]\rho}\hat{m}_\rho + \ell_5 u^{[\mu}\hat{M}^{\nu]\rho}\hat{m}_\rho \\
&+ \ell_6 u^{[\mu} M^{\nu]\rho}\hat{m}_\rho + \ell_7 \sigma^{[\mu}{}_\rho \hat{M}^{\nu]\rho} + \ell_8 \sigma^{[\mu}{}_\rho M^{\nu]\rho} + \ell_9 \hat{M}^{[\mu}{}_\rho M^{\nu]\rho} \\
&- T^{-2}S^{[\mu\nu]\rho}\left(a_\rho - \frac{1}{3}\Theta u_\rho\right) + T^{-2}\left(S^\rho{}_\rho{}^{[\mu}a^{\nu]} - \frac{1}{3}\Theta S^\rho{}_\rho{}^{[\mu}u^{\nu]}\right) + T_{BR}^{[\mu\nu]}
\end{aligned}
\tag{2}
$$

$$T^{-2}S^\mu{}_{\nu\rho} = 8\rho_0 u^\lambda M_{\nu\rho} + 2s_1 u^\lambda u_{[\nu}\hat{m}_{\rho]} + 2s_2 u^\lambda \hat{M}_{\nu\rho} + S_{BR}{}^\lambda{}_{\nu\rho}$$

with

$$
\begin{aligned}
T_{BR}^{\mu\nu} =& \frac{1}{2}\mathring{\nabla}_\lambda\left(S_{BR}{}^{\mu\nu\lambda} + S_{BR}{}^{\nu\mu\lambda} - S_{BR}{}^{\lambda\nu\mu}\right), \\
S_{BR}{}^\lambda{}_{\mu\nu} =& 2T^3\chi_1 \Delta^\lambda{}_{[\mu}u_{\nu]} + 2T^2\chi_2 M^\lambda{}_{[\mu}u_{\nu]} + 2\sigma_1 T^2 \sigma^\lambda{}_{[\mu}u_{\nu]} \\
&+ 2\sigma_2 T^2 \hat{M}^\lambda{}_{[\mu}u_{\nu]} + 2\sigma_3 T^2 \Delta^\lambda{}_\mu \hat{m}_{\nu]},
\end{aligned}
\tag{3}
$$

where we have decomposed the spin potential into transverse components,

$$\mu^{ab} = 2u^{[a}m^{b]} + M^{ab}, \tag{4}$$

with $m^a u_a = 0$ and $M^{ab}u_b = 0$ and defined

$$
\begin{aligned}
\Theta &= \mathring{\nabla}_\lambda u^\lambda, & a^\mu &= u^\alpha \mathring{\nabla}_\alpha u^\mu, \\
\Delta^{\mu\nu} &= g^{\mu\nu} + u^\mu u^\nu, & \Omega^{\mu\nu} &= \Delta^{\mu\alpha}\Delta^{\nu\beta}\mathring{\nabla}_{[\alpha}u_{\beta]}, \\
\sigma^{\mu\nu} &= \Delta^{\mu\alpha}\Delta^{\nu\beta}\mathring{\nabla}_{(\alpha}u_{\beta)} - \frac{1}{3}\Delta^{\mu\nu}\Theta,
\end{aligned}
\tag{5}
$$

which correspond to expansion, acceleration, the transverse projector, vorticity and the shear tensor, respectively. Calligraphic derivatives and hatted quantities are given by

$$
\begin{aligned}
\mathcal{D}_\alpha \sigma^{\alpha\beta} &= \mathring{\nabla}_\alpha \sigma^{\alpha\beta} - 3a_\alpha \sigma^{\alpha\beta}, \\
\mathcal{D}_\alpha \hat{M}^{\alpha\beta} &= \mathring{\nabla}_\alpha \hat{M}^{\alpha\beta} - a_\alpha \hat{M}^{\alpha\beta}, \\
\hat{m}^\mu &= m^\mu - a^\mu, \\
\hat{M}^{\mu\nu} &= M^{\mu\nu} + \Omega^{\mu\nu},
\end{aligned}
\tag{6}
$$

and circular brackets denote a symmetrized decomposition of indices, viz., $T^{(\mu\nu)} = \frac{1}{2}(T^{\mu\nu} + T^{\nu\mu})$. A more general expression for the constitutive relations for the energy momentum tensor and spin current can be found in the main text. See (18) and the ensuing discussion.

Adding terms of the form (3) to the stress tensor and current is usually referred to as a Belinfante-Rosenfeld transformation. Such terms will not modify the equations of motion and are often used to generate a symmetric stress tensor and vanishing spin current from an asymmetric stress tensor and its associated spin current. As we will see shortly, such terms should not be removed. While they do not contribute to the equations of motion, they do contribute to the expectation value of the stress tensor and current. This was first remarked on in [13].

Inserting (2) into (1) one obtains dynamical equations for the hydrodynamic variables $T$, $u^\mu$ and $\mu_{ab}$ which can be solved for once supplemented by initial conditions. At late times we expect that the system reaches thermodynamic (or hydrostatic) equilibrium where the temperature, velocity field, and spin chemical potential are fixed in terms of external forces acting on the system. In particular, we find that in equilibrium and in the absence of torsion, the

equilibrated spin chemical potential will be proportional to the thermal vorticity, $\Omega^{\mu\nu} - 2u^{[\mu}a^{\nu]}$ as predicted in [13, 14].

Spin current hydrodynamics may be relevant for the study of hyperon polarization measurements in heavy ion collisions. The prediction of global spin polarization in heavy-ion collisions, based on perturbative QCD, was initiated in [15, 16]. The first attempt to relate spin polarization to hydrostatic vorticity can be found in [17] and [18] and has been elaborated on in subsequent work [19–25]. See also [24, 26, 27] for reviews. This thread was continued by studies of entropy production in [28] and a classification of spin current sources and hydrodynamic constitutive relations in [29]. The latter work also pioneered a holographic study of spin transport. Other descriptions of spin hydrodynamics can be found in [30–33]. Despite these developments, a fully consistent set of constitutive relations which we derive in this paper did not appear in previous literature.

Before attempting to apply hydrodynamics with spin currents to the study of heavy ion collisions there are two issues that need to be addressed. The first, is whether such a hydrodynamic description is at all relevant for the physical process at hand, and the second is whether the hydrodynamical description of spin currents is complete. In this work we have addressed the second issue but not the first. Nevertheless, in section 4 we carry out a simple analysis of perturbed Bjorken flow associated with the constitutive relations (2). Using some coarse approximations we find a simple one parameter model that nicely fits the experimental results for hyperon polarization. See figure 1.

## 2   Spin current hydrostatics

In the presence of time independent sources such as an external metric or gauge field, the fluid is expected to reach a time independent hydrostatic equilibrium configuration whereby Euclidean correlators of the theory decay exponentially. This exponential decay implies that momentum space correlation functions at zero frequency are analytic in the spatial momenta implying that their associated generating function will be a local function of the background fields. Such a generating function was computed explicitly in [6, 7]. In what follows we use the same technique to study hydrostatically equilibrated spin current dynamics.

The sources which couple to the energy momentum tensor and spin current are the vielbein $e^a{}_\mu$ and spin connection $\omega_\mu{}^{ab}$,

$$\delta S = \int d^4x |e| \left( T^\mu{}_a \delta e^a{}_\mu + \frac{1}{2} S^\lambda{}_{ab} \delta \omega_\lambda{}^{ab} \right), \tag{7}$$

where the integral is over all space dimensions and a compact Euclidean time direction with parametric length $T_0^{-1}$. In a hydrostatic setting the sources will be time independent, viz.

$$\pounds_V e^a{}_\mu = 0, \qquad \pounds_V \omega_\mu^{ab} = 0, \tag{8}$$

where $V^\mu$ points in the time direction and $\pounds_V$ denotes its associated Lie derivative (the first equality implies that $V^\mu$ is a timelike Killing vector). The generating function for hydrodynamics with a spin current will be given by a local diffeomorphism and Lorentz invariant expression constructed out of the sources $e^a{}_\mu$ and $\omega_\mu{}^{ab}$, and the time direction $V^\mu$.

With some prescience let us denote

$$T = \frac{T_0}{\sqrt{-V^2}}, \qquad u^\mu = \frac{V^\mu}{\sqrt{-V^2}}, \qquad \mu^{ab} = \frac{\omega_\mu{}^{ab} V^\mu}{\sqrt{-V^2}}. \tag{9}$$

These quantities will correspond to the hydrostatic temperature, velocity field and spin potential respectively. To see this we consider the most general generating function which will lead

to constitutive relations which contain no derivatives of the parameters (9),

$$\ln Z_{\text{id}} = W_{\text{id}} = \int d^4x |e| P(T, M^2, m \cdot \tilde{M}, m^2), \tag{10}$$

where $M^2 = 2M_{\mu\nu}M^{\mu\nu}$ and $m \cdot \tilde{M} = m_\alpha M_{\beta\gamma} u_\delta \epsilon^{\alpha\beta\gamma\delta}$. We will refer to a fluid whose constitutive relations are completely determined by (10) as an ideal fluid.

The current and stress tensor associated with (10) are given by

$$T_{\text{id}}^{\alpha\beta} = \epsilon u^\alpha u^\beta + P\Delta^{\alpha\beta} - 2\left(\frac{\partial P}{\partial m^2} + 4\frac{\partial P}{\partial M^2}\right)u^\alpha M^{\beta\gamma}m_\gamma,$$
$$S_{\text{id}\,\alpha\beta}^\lambda = u^\lambda \rho_{\alpha\beta}, \tag{11a}$$

with

$$\epsilon = -P + \frac{\partial P}{\partial T}T + \frac{1}{2}\rho_{ab}\mu^{ab},$$
$$\rho_{\alpha\beta} = 8\frac{\partial P}{\partial M^2}M_{\alpha\beta} + \frac{\partial P}{\partial m \cdot \tilde{M}}\left(4\tilde{m}_{\alpha\beta} - u_\alpha\tilde{M}_\beta + \tilde{M}_\alpha u_\beta\right) \tag{11b}$$
$$- 2\frac{\partial P}{\partial m^2}\left(u_\alpha m_\beta - m_\alpha u_\beta\right),$$

once we restrict ourselves to a flat, torsionless geometry. Here we defined $\tilde{m}_{\alpha\beta} = -1/2\epsilon_{\alpha\beta\gamma\delta}u^\gamma m^\delta$.

There are several lessons to be learnt from (8) through (11). First, note that the identifications (9) yield the expected Gibbs Duhem relations once we identify $P$ with the pressure and $s = \partial P/\partial T$ with the entropy density $s$. As we will show in the next section, the entropy current $J^\mu = su^\mu$ is conserved for the ideal fluid, once the equations of motion are satisfied.

Second, since all sources are time independent, we find that (9) imply the hydrostatic relations

$$u^\mu K_\mu{}^{ab} = \mu^{ab} + e^a{}_\mu e^b{}_\nu\left(\Omega^{\mu\nu} - 2u^{[\mu}a^{\nu]}\right),$$
$$T\mathring{\nabla}_\lambda\frac{\mu^{\rho\sigma}}{T} = R^{\rho\sigma}{}_{\lambda\alpha}u^\alpha - 2K_{\lambda\alpha}{}^{[\rho}\mu^{\sigma]\alpha}, \tag{12}$$
$$a_\mu = -\frac{\mathring{\nabla}_\mu T}{T},$$

where $R$ is the Riemann tensor in the presence of torsion that is expressed in terms of the torsionless Riemann tensor as

$$R^{\rho\sigma}{}_{\lambda\alpha} = \mathring{R}^{\rho\sigma}{}_{\lambda\alpha} + 2\mathring{\nabla}_{[\lambda}K_{\alpha]}{}^{\rho\sigma} - 2K_\lambda{}^{\kappa[\rho}K_{\alpha\kappa}{}^{\sigma]}. \tag{13}$$

The first equality in (12) has been mentioned in [21,34] in the absence of torsion. It implies that a non vanishing spin potential must be supported by fluid vorticity or by acceleration (or, alternatively, temperature gradients) in order to maintain thermal equilibrium. In the absence of torsion, a non flat metric, or other external forces, the fluid will eventually settle down to a thermally equilibrated steady state in which the velocity field and temperature are covariantly constant. The first equality in (12) implies that the spin potential must vanish in such an equilibrated state. Therefore, if we wish to construct a gradient expansion around an equilibrium configuration we must count the spin potential as first order in derivatives.

Classifying the spin potential as a first order in derivatives term implies that the timelike components of the torsion tensor, $k^{ab}$, are also first order in derivatives. It remains to classify the transverse components of torsion, $\kappa_\nu{}^{ab} = \Delta^\mu{}_\nu K^\nu{}_{ab}$. In what follows we consider $\kappa_\mu{}^{ab}$ as

first order in derivatives, but it should be possible to set $\kappa_\mu{}^{ab}$ to be zeroth order in derivatives yielding torsio-hydrodynamics, an analog of magneto-hydrodynamics.

Before proceeding with higher order corrections to the ideal fluid, we remark that (12) implies that in the absence of torsion, $M^{\mu\nu} + \Omega^{\mu\nu} = 0$ and $m^\mu - a^\mu = 0$. Thus, there is an ambiguity in determining the constitutive relations (11). Of course, such an ambiguity will be resolved once non hydrostatic corrections are taken into account. In what follows we will consistently choose $M^{\mu\nu}$ and $m^\mu$ as hydrostatic variables in the absence of torsion over $\Omega^{\mu\nu}$ and $a^\mu$.

The hydrostatic gradient corrections to the constitutive relations can be obtained by expanding the hydrostatic generating function, $W$, in a derivative expansion. In this work we are interested in the equations of motion expanded to second order in derivatives. Since the antisymmetric components of the stress tensor sources the divergence of the spin current, we must expand the constitutive relations associated with the antisymmetric component of the stress tensor to second order in derivatives and the remaining constitutive relations to first order in derivatives. Therefore, to compute all possible corrections to the ideal fluid constitutive relations, we must classify all possible first order in derivative scalars which can contribute to the hydrostatic generating function, $W$, and all possible second order in derivative scalars which can contribute to the antisymmetric components of the stress tensor. In order to limit the number of such terms and also simplify future expressions we assume that the system is invariant under parity and also conformally invariant. A full analysis of the constitutive relations which are not restricted by symmetry will be discussed in a future paper.

The Weyl transformation of the spin connection associated with the Christoffel connection, $\mathring{\omega}_\mu{}^{ab}$ can be determined from the Weyl rescaling of the vielbein, $e^a{}_\mu \to e^\phi e^a{}_\mu$. In what follows we will assume that the spin connection transforms in the same way as $\mathring{\omega}_\mu{}^{ab}$. Alternately, that the contorsion tensor is inert under Weyl rescalings of the metric. One can argue that if the contorsion tensor transforms non trivially under Weyl rescalings then its transformation properties are such that a vanishing contorsion tensor is conformally equivalent to a non vanishing one [35, 36].

Using (1) we find that the change in the stress tensor and spin current due to an infinitesimal Weyl rescaling is given by

$$
\begin{aligned}
\delta T^{\mu\nu} &= -6\phi T^{\mu\nu} - S^{\mu\nu\rho}\partial_\rho\phi + S_\lambda{}^{\lambda\mu}\partial^\nu\phi\,, \\
\delta S^{\lambda\mu\nu} &= -6\phi S^{\lambda\mu\nu}.
\end{aligned}
\tag{14a}
$$

Using (7) we find that tracelessness is replaced by

$$
T^\mu{}_\mu = \mathring{\nabla}_\mu S_\lambda{}^{\lambda\mu}\,.
\tag{14b}
$$

We defer an extensive discussion of conformal invariance in the presence of torsion, and the recovery of the canonical transformation laws for the stress tensor in its absence to future work.

It follows that the transverse part of the spin potential, $M^{\mu\nu}$, transforms homogenously under Weyl rescalings while $m^\alpha$ does not. Thus, in a conformally invariant theory the pressure $P$ in (10) can depend only on $T$ and $M^2$. Counting $M^2$ as second order in derivatives implies that

$$
\epsilon = \epsilon_0 T^4, \quad P = \frac{1}{3}\epsilon_0 T^4, \quad \frac{\partial P}{\partial M^2} = \rho_0 T^2\,,
\tag{15}
$$

up to second order in derivative corrections.

It is now straightforward, though somewhat tedious to argue that the most general correction to $W$, $W_h$, at the order we are interested in is given by

$$
W_h = \int d^4x |e| \left( \chi^{(1)} T^3 \kappa + 2\chi_1^{(2)} T^2 \kappa_A^{\mu\nu} M_{\mu\nu} + 2\chi_2^{(2)} T^2 K^{\mu\nu} M_{\mu\nu} \right),
\tag{16}
$$

where $\chi^{(1)}$ and the $\chi_j^{(2)}$'s are numbers, $\kappa = u_a e^\mu{}_b K_\mu{}^{ba}$, $\kappa_A^{\alpha\nu} = u_\beta \Delta^{\mu[\nu} e^{\alpha]}{}_a K_\mu^{ab} e^\beta{}_b$ and $K^{\alpha\beta} = u^\mu \Delta^\alpha{}_\gamma \Delta^\beta{}_\delta e^\gamma{}_c e^\delta{}_d K_\mu{}^{cd}$. The stress tensor and spin current derived from (16) are given by

$$
\begin{aligned}
T_h^{(\mu\nu)} &= 4T^3 \chi^{(1)} u^{(\mu} m^{\nu)} + \mathcal{O}(\partial^2), \\
T_h^{[\mu\nu]} &= T^3 \chi^{(1)} \left(2u^{[\mu} m^{\nu]} - M^{\mu\nu}\right) + 4T^2 \left(\chi_1^{(2)} - \chi_2^{(2)}\right) u^{[\mu} M^{\nu]\alpha} m_\alpha + 2T^2 \chi_1^{(2)} \mathring{\nabla}_\alpha M^{\alpha[\nu} u^{\mu]}, \quad (17) \\
S_{h\,ab}^\lambda &= 2T^3 \chi^{(1)} \Delta^\lambda{}_{[a} u_{b]} - 4T^2 \chi_1^{(2)} M^\lambda{}_{[a} u_{b]} + 4T^2 \chi_2^{(2)} u^\lambda M_{ab},
\end{aligned}
$$

once we set the torsion to zero.

The contribution of the term associated with $\chi_2^{(2)}$ to the constitutive relations is identical to that of an ideal, conformal fluid at second order in the derivative expansion. At least as far as the antisymmetric part of the stress tensor and the spin current are concerned. Therefore, we may, without loss of generality remove the former by an appropriate shift of the latter.

## 3 Spin current hydrodynamics

The remaining contributions to the stress tensor and current, $T_r^{\mu\nu}$ and $S_{r\,ab}^\lambda$ contain all possible expressions which vanish in equilibrium, are parity invariant, and satisfy (14). We find

$$
\begin{aligned}
T^{-3} T_r^{(\mu\nu)} &= -\sigma_6 \sigma^{\mu\nu} + \left(\sigma_7 - 4\chi^{(1)}\right) u^{(\mu} \hat{m}^{\nu)} - \frac{1}{3} \chi^{(1)} \Theta \Delta^{\mu\nu} - \chi^{(1)} \Theta u^\mu u^\nu, \\
T^{-2} T_r^{[\mu\nu]} &= T\left(\sigma_7 - 2\chi^{(1)}\right) u^{[\mu} \hat{m}^{\nu]} + T\sigma_8 \hat{M}^{\mu\nu} + \Delta_\beta{}^{[\mu} u^{\nu]}\left(\lambda_1 \mathcal{D}_\alpha \sigma^{\alpha\beta} + \lambda_2 \mathcal{D}_\alpha \hat{M}^{\alpha\beta}\right) \\
&\quad + \lambda_3 \Delta^{\rho[\mu} \Delta^{\nu]\sigma} \mathring{\nabla}_\rho \hat{m}_\sigma + \lambda_4 u^{[\mu} \sigma^{\nu]\rho} \hat{m}_\rho + \lambda_5 u^{[\mu} \hat{M}^{\nu]\rho} \hat{m}_\rho \\
&\quad + (\lambda_6 - 4\chi_1^{(2)} + 8\rho_0) u^{[\mu} M^{\nu]\rho} \hat{m}_\rho + \lambda_7 \sigma^{[\mu}{}_\rho \hat{M}^{\nu]\rho} + \lambda_8 \sigma^{[\mu}{}_\rho M^{\nu]\rho} \\
&\quad + \lambda_9 \hat{M}^{[\mu}{}_\rho M^{\nu]\rho} + \frac{2}{3} \chi_1^{(2)} M^{\mu\nu} \Theta - T^{-2} S_r^{[\mu\nu]\rho}\left(a_\rho - \frac{1}{3}\Theta u_\rho\right) \\
&\quad + T^{-2}\left(S_r{}^\rho{}_\rho{}^{[\mu} a^{\nu]} - \frac{1}{3}\Theta S_r{}^\rho{}_\rho{}^{[\mu} u^{\nu]}\right), \\
T^{-2} S_{r\,ab}^\lambda &= 2\sigma_1 \sigma^\lambda{}_{[a} u_{b]} + 2\sigma_2 \hat{M}^\lambda{}_{[a} u_{b]} + 2\sigma_3 \Delta^\lambda{}_{[a} \hat{m}_{b]} + 2\sigma_4 u^\lambda u_{[a} \hat{m}_{b]} + 2\sigma_5 u^\lambda \hat{M}_{ab}.
\end{aligned} \quad (18)
$$

A few comments are in order. To help the reader identify the role of the various terms in (18) we have labeled coefficients associated with first order in derivative terms by $\sigma_i$ and coefficients associated with second order in derivative terms by $\lambda_i$. While one often denotes the shear viscosity by $\eta$ we have refrained from doing so for reasons that will become clear shortly. The $\chi^{(1)}$ and $\chi_i^{(2)}$ dependent terms appearing in (18) have been introduced in order to ensure that (14) are satisfied out of equilibrium. The same goes for the last two terms on the right hand side of the expression for $T^{[\mu\nu]}$.

Also, we have written (18) in what is usually referred to as the Landau frame where $u^\mu$ is an eigenvector of the stress tensor with negative eigenvalue. Frame transformations offer an additional freedom in redefining the spin potential which we avoid using at this order in the derivative expansion. The hydrostatic stress tensor and spin current, $T_{\text{id}}^{\mu\nu} + T_h^{\mu\nu}$ are written in a hydrostatic frame which is more natural from the point of view of the hydrostatic partition function.

To further simplify (18) it is convenient to make the redefinitions

$$
\begin{aligned}
&\sigma_6 = 2\eta_0 + \chi^{(1)}, && \sigma_8 = \eta_1 + \chi^{(1)}, \\
&\lambda_1 = \ell_1 + \sigma_1, && \lambda_2 = \ell_2 + \sigma_2, \\
&\lambda_3 = \ell_3 + \sigma_3, && \lambda_4 = \ell_4 - \sigma_3, \\
&\lambda_5 = \ell_5 - \sigma_3, && \lambda_6 = \ell_6 + \sigma_3, \\
&\lambda_7 = \ell_7 - \sigma_1 + \sigma_2, && \lambda_8 = \ell_8 - 2\chi_1^{(2)} + \sigma_1, \\
&\lambda_9 = \ell_9 + 2\chi_1^{(2)} - \sigma_2.
\end{aligned}
\tag{19}
$$

With these redefinitions, the terms in the spin current associated with $\chi^{(1)}$, $\chi_1^{(2)}$, and $\sigma_i$ with $i = 1, \ldots, 3$ reduce to Belinfante Rosenfeld terms, discussed in the introduction, and therefore do not affect the equations of motion.

As we have stressed earlier, the constitutive relations for the antisymmetric components of the stress tensor at order $n$ contribute to the equations of motion at order $n$ through the equation of motion for the spin current. This behaviour may be contrasted with the constitutive relations for the spin current at order $n$, or other components of the stress tensor at order $n$, which contribute to the equations of motion at order $n + 1$. This implies that the constitutive relations associated with the coefficients $\sigma_7$ and $\eta_1$ contribute to the equations of motion associated with the zero order in derivative constitutive relations for the current. Since $\chi^{(1)}$ doesn't contribute to the equations of motion, this implies that non vanishing $\sigma_7$ and $\eta_1$ yield the leading order equations of motion $\hat{m}^\mu = 0$ and $\hat{M}^{\mu\nu} = 0$ at leading order in a derivative expansion. That is, the hydrostatic values for $\hat{m}^\mu$ and $\hat{M}^{\mu\nu}$ are valid at least up to second order in derivatives. Thus, in a sense, dynamical spin can be obtained only for vanishing $\sigma_7$ and $\eta_1$. In the remainder of this work we will consider the dynamical spin limit which states that

$$
\sigma_7 = 0, \qquad \eta_1 = 0.
\tag{20}
$$

We discuss this limit further in section 5.

Combining (11), (17), (18) and (19), removing $\chi_2^{(2)}$ following the discussion after (17), setting $\sigma_7 = 0$ and $\eta_1 = 0$, and slightly relabeling coefficients, yields (2).

We note in passing that the terms associated with $\sigma_4$ and $\sigma_5$ can also be packaged as a Belinfante Rosenfeld term by adding a $\lambda_{10} T^2 u^\lambda u^{[\mu} \mathcal{D}_\lambda \hat{m}^{\nu]}$ and a $\lambda_{11} T^2 u^\lambda \mathcal{D}_\lambda \hat{M}^{\mu\nu}$ term to the antisymmetric part of the stress tensor and then redefining $\lambda_{10} = \ell_{10} + \frac{1}{2}\sigma_4$ and $\lambda_{11} = \ell_{11} + \sigma_5$ (with $u^\lambda \mathcal{D}_\lambda \hat{m}^\mu = u^\lambda \mathring{\nabla}_\lambda + \frac{1}{3}\theta \hat{m}^\mu - m \cdot \hat{m} + \hat{m} \cdot \hat{m} u^\mu$ and $u^\lambda \mathcal{D}_\lambda \hat{M}^{\mu\nu} = u^\lambda \mathring{\nabla}_\lambda \hat{M}^{\mu\nu} + \frac{1}{3}\theta \hat{M}^{\mu\nu} + 2u^{[\mu} \hat{M}^{\nu]\lambda} m_\lambda - 2u^{[\mu} \hat{M}^{\nu]\lambda} \hat{m}_\lambda$). The reason these last two terms don't appear in (18) is that we have substituted those expressions with their values under the equations of motion.

The various coefficients multiplying the tensor structures in (2), e.g., $\eta_0$, are restricted by positivity of entropy production, unitarity of retarded correlation functions or unitarity of the Schwinger-Keldysh generating function [9–12]. In what follows we will study restrictions on the coefficients in (2) coming from positivity of entropy production. A study of the restrictions on coefficients via other methods is left for future work.

Following [37] we posit the existence of an entropy current $J_S^\mu$ satisfying $\mathring{\nabla}_\mu J_S^\mu \geq 0$ under the equations of motion, such that for an ideal fluid $J_S^\mu = s u^\mu$ with $s = \partial P / \partial T$. For a non ideal fluid we take $J_S^\mu = s u^\mu + \mathcal{O}(\partial)$ where $\mathcal{O}(\partial)$ denotes corrections to the entropy current coming from explicit derivative terms appearing in the constitutive relations. Thus, the most general entropy current we may construct, to first order in derivatives is given by

$$
J_S^\mu = J_c^\mu + (s_1 \Theta u^\mu + s_2 a^\mu + s_3 m^\mu) T^2,
\tag{21}
$$

where

$$J^\mu_c = su^\mu - \frac{u_\nu}{T}\left(T^{\mu\nu} - T^{\mu\nu}_{\rm id}\right) - \frac{1}{2}\frac{\mu^{ab}}{T}\left(S^\mu{}_{ab} - S^\mu_{{\rm id}ab}\right) \tag{22}$$

is referred to as the canonical part of the entropy current. In a conformal theory the $s_i$ are constant.

When expanding the entropy current to first order in derivatives, the divergence of the entropy current is a second order in derivatives scalar. It is useful to classify the latter into two categories. The first are independent second order scalars, these are scalars which can not be written as products of first order scalars. The second includes products of first order scalars. All independent second order scalars appearing in the divergence of the entropy current must vanish on account of the positivity condition. For the same reason all products of first order scalars must arrange themselves into complete squares or vanish.

It is straightforward to show that

$$\begin{aligned}
\mathring{\nabla}_\mu J^\mu_c = &- \mathring{\nabla}_\mu\left(\frac{u_\nu}{T}\right)\left(T^{\mu\nu} - T^{\mu\nu}_{\rm id}\right)\\
&- \frac{1}{2}\mathring{\nabla}_\mu\left(\frac{\mu^{ab}}{T}\right)\left(S^\mu{}_{ab} - S^\mu_{{\rm id}ab}\right) - \frac{\mu^{ab}}{T}T_{ab}\,.
\end{aligned} \tag{23}$$

Inserting (21) into $\mathring{\nabla}_\mu J^\mu_S \geq 0$ and using (23) we find

$$s_1 = -\chi_1\,, \quad s_2 = \chi_1\,, \quad s_3 = -\chi_1\,, \quad \eta_0 \geq 0\,. \tag{24}$$

Let us make the following remarks. Since the spin potential is first order in derivatives the $n-1$th order spin current contributes to the $n$th order entropy current. Thus, the first order entropy current can only constrain the first order energy momentum tensor and zeroth order spin current. In practice, it constrains only $\eta_0$, the shear viscosity.

To determine constraints on the first order terms in the spin current one would need to go to second order in the entropy current. While we have not carried out such an analysis, we note that, at least for spin-less charged fluids, all constraints from the entropy current which imply equality type relations among transport coefficient are already implemented from the partition function. Further, all inequality type constraints appear at leading order in the entropy current [8].

Another somewhat unusual feature of hydrodynamics with a spin current is that the coefficient of the shear term in $T^{\mu\nu}$ is $-(\eta_0 + \chi_1)T^3$, c.f, (19). Nevertheless, it is $\eta_0$ that is constrained to be positive which is perhaps compatible with the fact that $\chi_1$ does not enter into the equations of motion. We have checked that positivity of $\eta_0$ also follows from positivity of the appropriate stress tensor correlator. Note that a computation of two point functions of the stress tensor require knowledge of the expectation value of the stress tensor in the presence of a background metric and spin connection which we have not presented here.

## 4 An application to heavy ion collisions

In this short letter we do not presume to carry out a full fledged analysis of heavy ion collision experiments with possible spin currents manifesting during the short collision period. Instead, we consider a perturbed solution to the hydrodynamic equations of motion in the presence of spin, in the limit of dynamical spin, with an underlying Bjorken ($SO(1,1) \times ISO(2) \times Z_2$) symmetry [38]. We then attempt to relate the dependence of the spin potential on the initial temperature to the dependence of the average hyperon polarization vector on the beam energy. Of course, a complete analysis, which we do not carry out in this short letter, should include a

proper treatment of initial conditions, a full hydrodynamic simulation, and a comprehensive treatment of hadronization of the quark gluon plasma before reaching the detector.

Consider a collision of two gold ions of radii $R$ initially moving with a relativistic velocity directed along a Cartesian '$z$' coordinate. Let's assume that the fluid formed after the collision has Bjorken symmetry, that is, it is invariant under boosts along the beam direction, translations and rotations along the '$x$' and '$y$' directions, and under $z/t \to -z/t$. Going to a Milne coordinate system, $ds^2 = -d\tau^2 + \tau^2 d\eta^2 + dx^2 + dy^2$ where $\tau = \sqrt{t^2 - z^2}$ and $\eta = \text{arctanh}(z/t)$ are proper time and pseudo-rapidity respectively, we find that

$$u^\tau = 1, \qquad T = T_0 \left(\frac{\tau_0}{\tau}\right)^{\frac{1}{3}} - \frac{\eta_0}{2\epsilon_0 \tau}, \tag{25}$$

with all other components of $u^\mu$ and $\mu_{ab}$ vanishing, solve the equations of motion. Here $T_0$ is the temperature at the initial time $\tau_0$ when the fluid description is a viable one. Note that the shear viscosity to entropy ratio, $\eta/s$, satisfies $\eta/s = 3\eta_0/4\epsilon_0$.

Since the spin potential vanishes on account of Bjorken symmetry, let's consider linear perturbations of Bjorken flow which break transverse translations and axial rotation, $T \to T + \int d^2q \delta T e^{i(q_x x + q_y y)}$, $u^\mu \to u^\mu + \int d^2q \delta u^\mu e^{i(q_x x + q_y y)}$, and $\mu_{ab} \to \mu_{ab} + \int d^2q \delta \mu_{ab} e^{i(q_x x + q_y y)}$. To mimic the experiment, we consider a peripheral collision with impact parameter $b$ along the '$x$' axis. Glancing beams are expected to create a non-trivial velocity gradient in the $x$ direction at initial proper time $\tau_0$ at which we assume hydrodynamics becomes applicable. To this end, we consider an initial velocity profile where $\delta u^\eta(\tau_0) \propto b q_x$, and other components of the perturbations to the velocity vanish. As a result, we find that $\delta m^\eta$, $\delta M^{\eta x} = \delta M^\eta i q_x$ and $\delta M^{\eta y} = \delta M^\eta i q_y$ are non zero while the temperature perturbations and all other components of the spin potential vanish.

To solve the equations of motion we will use the Floerchinger-Wiedemann (FW) approximation [39], where $\frac{3\eta_0}{4\epsilon_0}\frac{1}{T\tau}$ is perturbatively small but $q^2\tau^2 \frac{3\eta_0}{4\epsilon_0}\frac{1}{T\tau}$ (with $q^2 = q_x^2 + q_y^2$) is finite. In this approximation, only the leading term for the temperature in (25) becomes relevant, the velocity field perturbations take the form

$$\delta u^\eta = i u_0 \, b \, q_x \, \tau^{-\frac{5}{3}} e^{-\frac{9q^2 \eta_0 \tau_0}{16 T_0 \epsilon_0}\left(\frac{\tau}{\tau_0}\right)^{\frac{4}{3}}}, \tag{26}$$

and $\delta M^\eta$ and $\delta m^\eta$ are determined algebraically from $\delta u^\eta$ and its derivatives.

Presumably, the stress tensor and spin current will evolve according to hydrodynamic theory from an initial Bjorken time $\tau_0$ to a final time $\tau_f$ when matter hadronizes, $T(\tau_f) = T_f \simeq 150 MeV$. The hadrons yield is then collected by the detector which measures its properties. Converting a hydrodynamics spin current and energy momentum tensor to a Hadron distribution is fraught with difficulty. One often used prescription for doing so works under the assumption that the particle distribution after hadronization follows a thermal distribution with temperature, velocity and chemical potential of the hydrodynamic configuration leading to it [40]. Within this framework the polarization vector reads

$$\Pi_\alpha(p) = -\frac{1}{4}\epsilon_{\alpha\rho\sigma\beta}\frac{p^\beta}{m}\frac{\int d\Sigma_\lambda p^\lambda B \mu^{\rho\sigma}}{2\int d\Sigma_\lambda p^\lambda n_F}, \tag{27}$$

where $\int d\Sigma_\mu$ is an integral over the hadronization surface, $d\Sigma_\mu = \tau \delta^\tau_\mu d\eta dx dy$ in Bjorken coordinates, $p^\mu$ is the particle momentum, $m$ its mass, $n_F$ is the Fermi Dirac distribution and $B$ is an additional distributional quantity that depends on $u^\mu$ and $T$. See, e.g., [21, 41] for details.

It is tempting to use our solution to evaluate (27) and compare to data. However, one should keep in mind that our hydrodynamic solution is rather simple minded, involving a

linear perturbation on Bjorken symmetry on top of which we used the FW approximation. This perturbation should presumably capture a non vanishing impact parameter. Realistic collisions at mid centrality have an impact parameter of order of the nucleus size and are unlikely to resemble Bjorken flow. They should generate a large enough vorticity for a non trivial spin current to be generated which makes the validity of our linearized approximation somewhat suspect. Still, we have at our disposal an analytic solution to the hydrodynamic equations of motion with spin and it is hard to resist the temptation to compare it with the experimental results using (27). Hence, throwing caution to the wind, and inserting the perturbed Bjorken solution into (27), we find

$$
\Pi_\mu(p) = \frac{16 b e^{-\frac{4T_f^4 \epsilon_0 x_0^2}{9T_0^3 \eta_0 \tau_0}} u_0 \pi^{\frac{3}{2}} T_f^8 x_0 \epsilon_0^{\frac{3}{2}} (\ell_1 + \ell_2) \mathrm{Erf}\left(\sqrt{\frac{4T_f^4 \epsilon_0 y_0^2}{9T_0^3 \eta_0 \tau_0}}\right)}{27 m T_0^{\frac{13}{2}} \eta_0^{\frac{3}{2}} \ell_2 \tau_0^{\frac{13}{6}}} \times \begin{pmatrix} -p^y I^{(1)} \\ 0 \\ 0 \\ I^{(2)} \end{pmatrix}, \tag{28}
$$

where now $x_0 = R - \frac{b}{2}$, $y_0 = \sqrt{R^2 - \frac{b^2}{4}}$ and we have integrated over the range $-x_0 < x < x_0$ and $-y_0 < y < y_0$ which approximates the area of overlap of the two colliding nuclei. The expressions for $I^{(n)}$ are given by

$$
I^{(n)}(\tau_f) = \frac{\int d\eta \, B \, (p^\tau)^n}{2x_0 \times 2y_0 \times \int d\eta \, n_F p^\tau}\bigg|_{\tau = \tau_f}, \tag{29}
$$

and Erf denotes the error function.

We are particularly interested in the dependence of the spin polarization on the initial temperature, related to the beam energy. Making the reckless approximation that energy and nucleons are distributed uniformly in the nucleus and that the relation between energy density and temperature is of the form $\epsilon = \epsilon_0 T^4$ as dictated by conformal invariance, we find

$$
T_0 = \left(\frac{2N}{\pi R^2 \epsilon_0 \tau_0}\right)^{\frac{1}{4}} s_{NN}^{\frac{1}{8}}, \tag{30}
$$

where $N$ is the number of nucleons, $\sqrt{s_{NN}}$ is the beam energy per nucleon, and we approximated the volume of the nucleus as $\pi R^2 \tau_0$. It is clear that each of these approximations may be improved and upgraded, but as a preliminary order of magnitude estimate relating our hydrodynamic solution to the polarization vector, they are good enough.

Using, $\epsilon_0 = 12$, $T_f = 150 MeV$, $\eta/s = 1/4\pi$, $\tau_0 = 1 fm$, $R = 7 fm$ and $b = 10 fm$ (see [39, 42–44]), we find

$$
\frac{4T_f^4 \epsilon_0 x_0^2}{9T_0^3 \eta_0 \tau_0} \simeq \frac{5.1}{\left(\frac{s_{NN}}{\mathrm{GeV}^2}\right)^{\frac{3}{8}}}, \qquad \sqrt{\frac{4T_f^4 \epsilon_0 y_0^2}{9T_0^3 \eta_0 \tau_0}} \simeq \frac{5.5}{\left(\frac{s_{NN}}{\mathrm{GeV}^2}\right)^{\frac{3}{16}}}. \tag{31}
$$

The overall scaling of $\Pi$ in terms of the energy per nucleon is of the form

$$
\Pi = \alpha \frac{\exp\left(-\frac{5.1}{\left(\frac{s_{NN}}{\mathrm{GeV}^2}\right)^{\frac{3}{8}}}\right) \mathrm{Erf}\left(\frac{5.5}{\left(\frac{s_{NN}}{\mathrm{GeV}^2}\right)^{\frac{3}{16}}}\right)}{\left(\frac{s_{NN}}{\mathrm{GeV}^2}\right)^{\frac{13}{16}}}, \tag{32}
$$

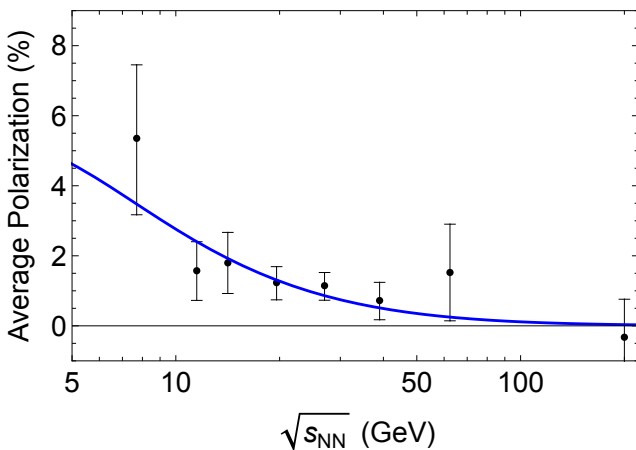

Figure 1: A comparison of our estimate (32) of the average hyperon polarization (blue) to the STAR measurement [1]. Since a magnetic field was not incorporated in our setting, we have compared our estimate to the average value of the polarization of $\Lambda$ and $\overline{\Lambda}$.

where we have replaced the overall constant in (28), which depends on the undetermined initial value for the velocity field perturbations $u_0$, and on the coefficients $\ell_1$ and $\ell_2$, with $\alpha$. To compare to experiment we need to work out $\Pi_\mu$ in the center of mass frame of the hyperon. Such a Lorentz transformation will not affect the dependence of $\Pi_\mu$ on $s_{NN}$. Therefore, we can attempt to fit (32) to experiment by fitting to a single parameter, $\alpha$. Using the data from [1], we find a surprisingly good fit to $\alpha = 286 \pm 52$. See figure 1. We emphasize that our phenomenological analysis crucially depends on the new transport coefficients $\ell_1$ and $\ell_2$ in the constitutive relations.

## 5   Discussion

In this paper, we initiated a fully fledged study of relativistic spin hydrodynamics. The hydrodynamic constitutive relations in the dynamical spin limit, relating the spin current and the stress tensor to fluid velocity, temperature, spin potentials and their derivatives, can be found in (2). The dynamical spin limit is one where the coefficients $\sigma_7$ and $\eta_1$ (c.f. (18)) associated with the antisymmetric components of the stress tensor at first order in derivatives have been set to zero. Setting transport coefficients to zero is somewhat unusual. Often, in order for transport to vanish there must be an underlying symmetry which ensures that the said coefficient is trivial, or there is a physical reason for the coefficient to be irrelevant to the dynamics. Apart from the surprisingly good fit of our model to data, as exhibited in figure 1, we have not found a reason for $\sigma_7$ and $\eta_1$ to vanish. We will explore the dynamics of systems with non vanishing $\sigma_7$ and $\eta_1$ in future work. For consistency, the constitutive relations for the symmetric part of the stress tensor and spin current were expanded to first order in derivatives while those for the antisymmetric part of the stress tensor were expanded to second order in derivatives. This mismatch in the derivative expansion is a result of the hydrostatic equilibrium relation (12) between torsion, spin potential, vorticity and acceleration which implies that the spin potential and the longitudinal component of the torsion must be first order in derivatives.

Regardless of the dynamical spin limit, we have for consistency, expanded the constitutive relations for the symmetric part of the stress tensor and spin current to first order in derivatives and for the antisymmetric part of the stress tensor to second order in derivatives. However, in certain condensed matter systems such as graphene torsion is used as the long-wavelength description of dislocations and disclinations in the atomic structure [45, 46]. In these systems a sensible choice might be to keep the spatial torsion at zeroth order in derivatives along with temperature and velocity. It would be interesting to develop such a torsio-hydrodynamic theory further, following the route we outlined here. Recent works [47–51] strongly suggest that graphene, as well as certain clean Dirac and Weyl semimetals, are well described by hydrodynamic theory which further motivates this study.

Another possible direction is to extend our results to non parity invariant and non-conformal theories. Indeed, Bayesian analysis of heavy-ion data suggest that bulk viscosity may [52, 53] or may not [54, 55] play an important role in the hydrodynamic description of heavy ion collisions. Finally, for off-central heavy ion collisions, which is the regime where spin hydrodynamics is most relevant, it is desirable to employ a more realistic hydrodynamic configuration in which the rotation symmetry around the beam axis is broken. In this context, extension of the recently introduced a hydrodynamic frame [56, 57] that allows a consistent set of hyperbolic equations, to include spin currents is also desirable.

## 6 Acknowledgements

We thank F. Becattini, K. Fukushima, G. Torrieri and J. Zaanen for useful discussions. DG and UG are partially supported by the Delta-Institute for Theoretical Physics (D-ITP) funded by the Dutch Ministry of Education, Culture and Science (OCW). In addition, DG is supported in part by CONACyT through the program Fomento, Desarrollo y Vinculacion de Recursos Humanos de Alto Nivel. AY is supported in part by an Israeli Science Foundation excellence center grant 2289/18 and a Binational Science Foundation grant 2016324.

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
