# Peer review of "Hydrodynamics of spin currents"

_SciPost Physics, doi:SciPost Phys. 11, 041 (2021)_

## Round 2 · Referee Report · Michal P. Heller (Referee 1) · 2021-5-29

Report

Hydrodynamics with spins is one of the emerging subjects in the topics of heavy-ion collisions -- the primary subject of the present manuscript -- with strong relevance also for condensed matter physics.

The authors provide a comprehensive analysis of hydrodynamic with spins from the point of view of effective field theory and characterize contributions to hydrodynamic constitutive relations up to and including first order in derivatives. Subsequently, they perform a phenomenological analysis and, despite using a rather simplistic model, they manage to fit experimental data upon fitting a single parameter, which I find impressive.

The paper is well written and comprehensive and I will be happy to endorse it for publication upon I hear from the authors their response on the following, mostly presentational points:

1) Introduction section contains many flashforwards to the results, so in my view it would be more appropriate to call it an introduction and summary.

2) I am not sure I get why the authors a dot over nabla to denote a covariant derivative. They use regular partial derivatives symbol and they I believe regular nabla would be enough. I am writing this because the paper is heavy in notation and steps to lighten it would be welcome.

3) The original Bjorken paper on the boost-invariant flow is not cited anywhere in the paper (might be relevant for the condensed matter audience, although the flow itself is well introduced in the paper).

4) Since by now there are many works on spin hydrodynamics and the paper aims to be comprehensive, it would be very useful for a reader (including myself) to see which parts/aspects of the constitutive relations the authors derive were already known and why are genuinely new.

5) Regarding the phenomenological analysis the authors did, it would be like-wisely interesting to know if such an analysis relies on the fact that they added some new terms to the constitutive relations that improve the agreement with the data or simpler constitutive relations would do or Bjorken flow + linearization are just sensitive to some parts of the constitutive relations.

6) In practical applications to heavy-ion collisions one would need to use equations of motions that are hyperbolic in nature akin to Israel-Steward formulation of relativistic viscous hydrodynamics or the more recent Bemfica-Disconzi-Noronha/Kovtun formulation. It would be useful if authors make this point apparent in their study.

  • validity: -
  • significance: -
  • originality: -
  • clarity: -
  • formatting: -
  • grammar: -

Author:  Umut Gursoy  on 2021-06-21  [id 1514]

(in reply to Report 1 by Michal P. Heller on 2021-05-29)

We thank the referee for a careful reading of the manuscript and insightful comments. We will respond to the comments in the same order as given by the referee:

  1. We will modify the title of the first section as requested.
  2. The difference between a covariant derivative with and without ring is that the former refers to derivative with Christoffel connection (no torsion) and the latter, one with torsion. The referee is right that we do not use the latter in this short paper as we expand expressions in torsion. However, we prefer to keep the notation with the ringed nabla in face of our upcoming works where we do need to make the distinction between the two clear. In particular, the torsionful derivative becomes highly useful in a hydrodynamic expansion where torsion is kept finite.
  3. We will add a reference to Bjorken's original paper.
  4. As far as we know the full consistent constitutive relations were not derived anywhere in the literature so the bulk of the results presented in the draft are new. We will add a sentence explaining this in the paragraph before the last in section I.
  5. Our phenomenological analysis crucially depend on the new terms added to the constitutive relations. In particular Bjorken + fluctuation would result in vanishing spin current without the transport coefficients $l_1$ and $l_2$. In this sense, these terms certainly affect the agreement with data. We will add a sentence in the last paragraph of section 4 to make this clear.
  6. We agree. We will add a sentence in the Discussion to address the referee's comment on the BDN/K frame.

We hope that we successfully addressed the comments of the referee with the aforementioned modifications (which will be listed in a separate accompaniment) and that our modified draft can be granted publication.

---

## Round 2 · Referee Report · Anonymous (Referee 2) · 2021-7-5

Report

The paper “hydrodynamics of spin currents” performs a study of the first-order theory of hydrodynamics that results from including spin currents in the effective gradient expansion of relativistic hydrodynamics. The study is well motivated from at least two points of view. Firstly it is always of interest to formulate effective descriptions of symmetry-allowed quantities, we never know when and in what context they become useful. Secondly, a potentially useful applications of this theory is in the context of heavy-ion physics and spin polarization (of e.g. the Lambda hyperon). Indeed the authors include a preliminary analysis of their hydrodynamic theory to this case of interest.

Technically, while involved, the paper follows the standard route of constructing such hydrodynamic theories, as described clearly in Landau and Lifshitz, and the results appear trustworthy. As the authors note themselves it would be desirable in the future to use perhaps more modern approaches (e.g. coupling to arbitrary backgrounds) in order to obtain potentially further constraints on the transport coefficients appearing in this work.

The only minor complaint I have (and I would make addressing this optional) is presentational and notational. Firstly, in a paper with as many symbols and indices as this one, I would prefer it if every symbol were defined right after its first appearance. As obvious as it might seem to the authors, this is for example not done for the spin current which first appears in Eq. (1) and is not formally identified until much later.

Secondly, I would recommend denoting the ‘ideal’ part of the stress tensor and the spin current not by the sub-script `i’, but in some other way (e.g {\rm id} or {\rm ideal}.) as it is too easily confused with one of the many indices.

Typo: below equation (7) the authors refer to ‘parameteric’, when I think they mean ‘parametric’

  • validity: top
  • significance: high
  • originality: high
  • clarity: high
  • formatting: perfect
  • grammar: perfect

Author:  Umut Gursoy  on 2021-07-22  [id 1605]

(in reply to Report 2 on 2021-07-05)

We thank the referee for the detailed reading and insightful comments. Indeed our paper is technical. While we agree with the referee that defining every symbol after its first appearance is helpful, we prefer not to do this with the spin current below Eq. (1) because it is clearly mentioned in the paragraph just above that equation. Instead, we went through the draft to check if there are any symbols left undefined and discovered one: $\tilde m_{ab} = -1/2 \epsilon_{abcd} u^c m^d$ in eq. (11b) which we define in the new version below that equation. We agree that using subscript "i" denoting ideal currents is confusing hence we replaced it with "id" throughout. We also corrected the typo mentioned by the referee.

---

## Round 3 · Referee Report · Michal P. Heller (Referee 1) · 2021-7-28

Report

The paper is good to go as it is.

---

## Round 3 · Author Response

This is resubmitted from v3 of the draft on the arXiv.

---

## Round 3 · List of Changes

1. Added reference to Bjorken's original paper.
  2. Added a sentence in the last paragraph of section 4 to explain that the the new transport coefficients play a crucial role in obtaining a nonvanishing polarization of Lambda hyperons.
  3. Added a sentence referring to extensions of the BDN/K hyrodynamics frame as possible outlook.
  4. Corrected a typo below equation (7) ‘parameteric’ to ‘parametric’
  5. Replaced subscript 'i' for the ideal currents with 'id' for clarity
  6. Defined $\tilde m_{ab}$ in eq. (11b) below it.
  7. Corrected a typo in Eq. (11b) and in (12).
  8. Added a new equation (13) to clarify our definitions.

---

## Editorial Decision

published